# Synbiotics and Gut Microbiota: New Perspectives in the Treatment of Type 2 Diabetes Mellitus

**DOI:** 10.3390/foods11162438

**Published:** 2022-08-13

**Authors:** Haoran Jiang, Miaomiao Cai, Boyuan Shen, Qiong Wang, Tongcun Zhang, Xiang Zhou

**Affiliations:** College of Life Sciences & Health, Wuhan University of Science & Technology, Wuhan 430065, China

**Keywords:** synbiotics, gut microbiota, type 2 diabetes mellitus, *Lactobacillus plantarum*

## Abstract

The number of people with type 2 diabetes mellitus (T2DM) has increased sharply over the past decades. Apart from genetic predisposition, which may cause some of the diagnosed cases, an unhealthy diet and lifestyle are incentive triggers of this global epidemic. Consumption of probiotics and prebiotics to gain health benefits has become increasingly accepted by the public in recent years, and their critical roles in alleviating T2DM symptoms are confirmed by accumulating studies. Microbiome research reveals gut colonization by probiotics and their impacts on the host, while oral intake of prebiotics may stimulate existing metabolisms in the colon. The use of synbiotics (a combination of prebiotics and probiotics) can thus show a synergistic effect on T2DM through modulating the gastrointestinal microenvironment. This review summarizes the research progress in the treatment of T2DM from the perspective of synbiotics and gut microbiota and provides a class of synbiotics which are composed of lactulose, arabinose, and *Lactobacillus plantarum*, and can effectively adjust the blood glucose, blood lipid, and body weight of T2DM patients to ideal levels.

## 1. Introduction

The incidence of diabetes mellitus is increasing every year with the improvement in living standards, and it has become another chronic disease that seriously endangers human health, apart from tumor and cardiovascular diseases. Data show that the number of adults with diabetes worldwide reached 537 million (~10.5%) in 2021, with an astonishing 16% increase compared to 2019. According to the International Diabetes Federation (IDF), this number will reach 783 million by 2045, and the proportion of adults endangered may reach one-eighth of the total adult population [1]. Among all sufferers of diabetes, the incidence of type 2 diabetes mellitus (T2DM) accounts for more than 90%. Diabetes can lead to serious health consequences such as hypertension, dyslipidemia, arteriosclerosis, and impaired renal function, and serious complications may occur such as blindness, stroke, myocardial infarction, and lower limb amputation. All these acute and chronic complications can affect the quality of patients’ lives to a certain extent. In the past, T2DM was a disease mainly found in the elderly, but increasing data show that T2DM is becoming more prevalent in young people, and up to 16% of the adult population worldwide with T2DM experienced adolescent onset of the disease [2].

Current drugs for the treatment of T2DM are insulin and a series of oral hypoglycemic agents, but T2DM is not easy to cure [3]. Because of insulin resistance and insulin deficiency, there are some limitations to past methods of treating T2DM. In recent years, much attention has been given to gut microbiota and their potential regulatory effects on T2DM. The role of intestinal flora in the development of T2DM has been revealed by further studies [4]. The digestive, absorptive, metabolic, and neuromodulatory functions of the intestine directly affect the metabolism of glucose and lipids, and the activities of gut microbiota play an important role in this process [5]. Therefore, the clinical efficacy of T2DM is closely related to abnormal changes in gut microbiota in patients. Synbiotics, which are composed of probiotics and prebiotics, can affect the efficacy of T2DM by regulating the balance of intestinal flora in patients with T2DM [6]. This review systematically elucidates the mechanisms of action and therapeutic effects of synbiotics and gut microbiota in the treatment of T2DM. Moreover, a class of synbiotic combinations consisting of lactulose, arabinose, and *Lactobacillus plantarum* are provided, which show a very promising effect on T2DM patients by modulating their gut microbiota.

## 2. The History of T2DM Treatment

T2DM is increasingly common worldwide since its discovery long ago. It is often accompanied by obesity, hypertension, dyslipidemia, arteriosclerosis, renal insufficiency, and other diseases, making it a major global health threat [7,8,9]. Patients with T2DM hardly notice the disease in its early stage, since at first it has few symptoms, such as mild fatigue and thirst, while elevated blood sugar can only be confirmed by a glucose tolerance test [10]. More than 40 antidiabetic drugs, including insulin and oral hypoglycemic drugs, have been approved for treatment since the 1920s [11].

Insulin is one of the main drugs for the treatment of diabetes. It was discovered and extracted in 1921 by Canadian scientists Banting and Best, and this was a milestone in the treatment of diabetes; however, repeated injections of insulin cause much inconvenience to patients. Meanwhile, insulin is by no means a panacea for diabetes [12,13]. In 1979, the International Diabetes Federation (IDF) proposed terms that have been used ever since: type 1 (i.e., insulin dependent), type 2 (i.e., noninsulin dependent), and “other” in cases that fit neither category [14]. Insulin has a prominent effect on type 1 diabetes with abnormal insulin secretion, which only accounts for approximately 10% of the total number of diabetic patients, mostly children and adolescents. On the contrary, insulin is much less effective in T2DM, which is far more common (more than 90%) than type 1 diabetes mellitus, and its main cause is progressively impaired insulin secretion by pancreatic β-cells, upon a pre-existing background of insulin resistance in skeletal muscle, liver, and adipose tissue [15].

Sulfonylureas, as another class of drugs for T2DM, were first discovered by French researchers, Janbon and Loubatiere, to show a hypoglycemic effect. In 1955, the first sulfonylurea drug, amisbutamide, was used in clinical research, but it was quickly withdrawn from the market due to the fact of its adverse side effects. Subsequently, the first-generation sulfonylurea drugs, including tolbutamide, chlorpropamide, acetaminophen, and tolazamide, were put on the market with mild hypoglycemic effects and serious adverse reactions such as liver function impairment. Since 1966, the second generation of sulfonylureas was discovered, and they have been widely used to date, becoming important drugs in the treatment of diabetes [16]. In 1969, second-generation sulfonylurea drugs were used in clinical trials and thereafter in patients. Compared to the first-generation drugs, they have the advantages of a strong hypoglycemic effect, few adverse reactions, and a low failure rate. The cardiovascular safety of sulfonylureas and their effects on β-cells are critical concerns, while extensive clinical studies are needed regarding these issues [17]. Another study, which analyzed 77 studies, showed that the use of sulfonylureas may increase the risk of cancer in patients with T2DM [18].

Metformin, an oral biguanide which is well established as a first-line treatment of T2DM, lowers blood glucose by promoting hepatic adenosine monophosphate kinase phosphorylation, which inhibits hepatic gluconeogenesis and, secondarily, decreases circulating insulin levels [19]. Its pharmacological effect is to increase the utilization of glucose by peripheral tissues, inhibit hepatic gluconeogenesis, and reduce glucose output and the absorption of glucose in the gastrointestinal tract, stimulating the hypothalamus to induce satiety and reduce intake, thereby enhancing fatty acid oxidation, improving lung mechanics, and reducing body weight, thus alleviating blood glucose levels and the burden of insulin secretion by the pancreas [20,21,22] (Figure 1). In addition, metformin has been shown to alter the composition of the gut microbiota, indicating that it interacts with the gut microbiota by modulating inflammation, glucose homeostasis, gut permeability, and short-chain fatty-acid-producing bacteria [23]. In patients with diabetes-related gut dysbiosis, metformin promotes the production of butyrate and propionate and improves a patient’s ability to break down amino acids [24]. Adverse reactions to metformin include lactic acidosis and renal and liver function damage [25]. There are many ongoing randomized trials investigating the effects of metformin on cancer-related outcomes, but it is unclear if metformin can exert direct effects on tumors in patients [19].

Other drugs, including α-glucosidase inhibitors, insulin sensitizers, and glinide insulin secretagogues, are also used in the treatment of T2DM. Although these drugs play a certain role in the treatment of T2DM, they inevitably have limitations [26]. For example, α-glucosidase inhibitors can cause abdominal distension, insulin sensitizers can increase the risk of myocardial infarction, and the high-frequency intake of glinides can make them inconvenient for patients to use [27,28].

Intriguingly, there is increasing evidence showing that the gut microbiota is closely related to the occurrence of T2DM. Diets supplemented with prebiotics or synbiotics may improve lipid metabolism and glucose homeostasis and enhance antioxidant enzyme activity in patients with T2DM [29,30]. Recently, an increasing number of researchers have been applying synbiotics in T2DM patients to improve their intestinal microenvironment, thereby improving the efficacy of T2DM treatment, which is a promising field that deserves extensive study. It is safer and more precise in the control of blood sugar in patients with T2DM, which is an incomparable advantage over traditional chemical drugs.

## 3. Definition, Function, and Study of Synbiotics

### 3.1. Definition

As a new generation of microecological regulators, synbiotics and prebiotics can exert the physiological functions of probiotics and prebiotics simultaneously, fighting diseases together while maintaining the microecological balance within the gastrointestinal tract. The concept of synbiotics was first defined in 1995 as “a mixture of probiotics and prebiotics, with beneficial effects on the host by improving survival and implantation of live microbial dietary supplements in the gastrointestinal tract, by selectively stimulating growth and/or by activating the metabolism of one or a limited number of healthy bacteria” [31]. This definition is essentially consistent with the definitions of prebiotics and probiotics, which need to be complementary and synergistic in their roles [32]. In May 2019, the International Society for Probiotic and Prebiotic Sciences (ISAPP) convened a panel of nutritionists, physiologists, and microbiologists to review the definition and scope of synbiotics. The team updated the definition of commensal microorganisms to “a mixture of matrices selectively utilized by living and host microorganisms, which is beneficial to the health of the host” [33].

### 3.2. Synbiotics—Components and Function

Probiotics are living microorganisms that, when properly consumed, can provide health benefits to the host. In recent years, the consumption of probiotics for the promotion of health and well-being has increased worldwide. However, there are conflicting clinical results for many probiotic strains and formulations [34]. Probiotics can maintain the best combination of intestinal flora through interaction with their metabolites. Animal experiments and clinical trials have proved that probiotics can lower blood lipids and cholesterol [35]. Moreover, studies have shown that probiotics can regulate blood pressure by active substances, mainly from antihypertensive peptides and peptidoglycan [36]. They also stimulate the immune system and improve the overall immune function [37].

Prebiotics are substrates that are selectively utilized by host microorganisms for health benefits. They stimulate the growth and/or metabolic activation of colonic microbials to maintain a healthy state of the host’s gut microbiota [38]. Their function is realized by promoting the growth and proliferation of intestinal beneficial bacteria, which can regulate the intestinal microecological balance, thereby improving homeostasis to maintain body health.

Synbiotics refer to a product containing both probiotics and prebiotics. Probiotics and prebiotics complement each other to exert a synergistic effect on the host [39]. Synbiotics benefit the host by selectively stimulating the growth of probiotics or by activating bacterial metabolic pathways that promote human health, increasing the survival rate of probiotics in the gastrointestinal tract and making the gut microenvironment healthier. Several studies indicate that synbiotics are beneficial to the gastrointestinal and urinary tract, and even show an antitumor and anti-aging effect. The term “synbiotics” reflects the synergistic relationship between the beneficial bacteria and their selective substrates to stimulate their growth when they survive passing through the stomach to the large intestines to establish their predominance [40]. As a microecological compound, synbiotics show a variety of effects on human health such as protecting the liver, regulating blood lipids, inhibiting tumors, and treating vaginitis and urinary tract infections, among which regulating the intestinal flora and moistening the intestines is the most direct and fundamental function of synbiotics [41].

### 3.3. Synbiotics in Disease Treatment

The colon is one of the most important places where synbiotics work [42]. As one of the most metabolically active organs of the human body, the colon harbors an extremely complex microbial ecosystem that not only serves as a barrier against infection but also plays an active role in recovering energy from indigestible food components that cannot be affected by human enzymes. The study of colonic microbiota has become increasingly intensive, especially with the use of newly developed molecular methods, which can bring great value to human beings. In past studies, two main methods have been used to control the gut microbiota: probiotics and prebiotics [22]. Later, researchers tried to combine the two and, thus, put forward the concept of and studies on synbiotics.

The earliest proposed strategy to modulate the gut microbiota was the use of live microorganisms. Probiotics, as we now call them, have been produced and used for more than a hundred years ago [32,43,44], although the true definition of probiotics came many years later. There are hundreds of probiotic strains and related products to treat various diseases and improve the intestinal microenvironment. Systematic reviews and meta-analyses show that probiotics contribute to improved lactose digestion [45], the treatment of antibiotic-related diarrhea [46]; prevention of necrotizing enterocolitis of prematurity [47]; induction of remission of inflammatory bowel disease [48]; prevention and control of hyperglycemia [49]; improvement in total cholesterol, high-density lipoprotein (HDL), and tumor necrosis factor (TNF)-alpha levels in patients with nonalcoholic fatty liver disease (NAFLD) [50]; decreases in blood glucose and insulin as well as a homeostasis model assessment of insulin resistance in diabetic patients [51]. Jonkers et al. also reported that probiotics are effective in preventing antibiotic-associated diarrhea and reducing the infectivity of respiratory tract infections [52].

However, the research history on prebiotics has not been as long as probiotics. Although prebiotics have been used for a long time, the concept of prebiotics was not formally proposed until 1995. At present, the most widely studied prebiotics are fructans and galactose, mainly as inulin, fructooligosaccharides (FOSs), and galactose (GOSs). Various other oligosaccharides have also been investigated including low isomaltose, sodium oligomannose, pectin, resistant starch, low xylose, arabinoxylan, and human milk and milk oligosaccharides [53]. Relevant studies have confirmed that prebiotics provide health benefits through several different mechanisms including changes in microbial composition or metabolism, stimulation of growth or activity of health-promoting bacteria, and production of short-chain fatty acids [54]. Short-chain fatty acids can reduce local pH, induce the production of immunoregulatory cytokines, and stimulate mucin production, thereby improving the intestinal microenvironment [55].

The study of synbiotics is based on the above two findings. Although the concept of synbiotics officially arrived in 1995, the study of synbiotics started long before. Tanaka et al. studied the effects of galactooligosaccharides (GOS) in combination with *Bifidobacterium-4006* on 16 healthy adult men in 1983, and the results showed that this synbiotic combination was able to increase levels of commensal bifidobacteria, whereas probiotics alone did not show the same effect [56]. Although this was a very early study and changes in the gut microbiota were assessed only by colony cultures, the selection of probiotics and their complementary prebiotics as well as an appropriate probiotic and prebiotic control allowed us to elucidate the mechanisms of synbiotic efficacy. Basically, synbiotics are capable of increasing the number of probiotics and bifidobacteria.

After the 1990s, researchers began to study various synbiotic combinations to explore their relationship with intestinal flora and the mechanism of action through synbiotics to improve homeostasis of the intestinal microenvironment for treating various diseases.

Rather than simply mixing probiotics and prebiotics, the researchers proposed the concept that probiotics and prebiotics require synergy. The components of synbiotics include probiotics and prebiotics, which can work together, and synbiotic preparations have the advantage of resolving the responder/nonresponder phenomenon. To play a role in the gastrointestinal tract, probiotics not only ensure the safety of nutrition and other growth factors but also compete with the gut microbiota. By providing a niche opportunity for probiotics in the form of selectively fermented probiotics, the strain has a competitive advantage. Therefore, it is possible to significantly enhance their competitiveness and enhance their durability [23]. This approach is consistent with the theory of invasive ecology, where an invasive species is successful when it can exploit new resources or overcome commensal species for these resources, thereby successfully colonizing and establishing themselves [57,58,59].

With more research on synbiotics, various synbiotic products have entered people’s lives. Different synbiotic combinations have positive effects in the treatment of many diseases including atopic dermatitis, gastrointestinal disorders, hepatic encephalopathy, irritable bowel syndrome, metabolic disorders, and T2DM [29,60,61,62,63,64]. As a new generation of products following probiotics and prebiotics, synbiotics show their common advantages and cooperate to achieve a better therapeutic effect.

## 4. Synbiotics in the Treatment of T2DM

In recent years, studies on synbiotics, the intestinal microenvironment, and T2DM have gradually accumulated. More studies have shown that synbiotics can exert positive effects on the control of T2DM, which is good news for patients with T2DM. At present, the pathogenesis of T2DM has not been fully elucidated. However, there is evidence that the composition of the gut microbiota is associated with the development of T2DM and that the gut microbiota affects the host through its effects on body weight, bile acid metabolism, proinflammatory activity, insulin resistance, and regulation of gut hormones. The use of synbiotics to modulate the gut microbiota may be beneficial in improving host glucose metabolism and insulin resistance [65]. There is a close relationship between T2DM and changes in the composition of the intestinal microbiota, and it has been found that the relative abundance of firmicutes is low, and the proportions of *Basilides* and Proteus are higher in patients with T2DM [66,67]. The intestinal microenvironment in patients with T2DM was significantly changed compared with that of normal subjects, suggesting that there is a link between the development of T2DM and the intestinal microenvironment.

At present, clinical treatments for T2DM mainly include drug control of blood glucose levels or insulin therapy. However, there are few studies promoting the therapeutic effect on patients of regulating the intestinal microbiota balance. The metabolism of glucose and lipids is closely related to the digestive and absorptive functions of the intestine, and the metabolic activities of gut microbiota play an irreplaceable role in this process. Previous studies have shown that blood glucose, lipids, body weight, and body functions, including autoimmunity, are directly linked to the balance of the gut microecology in patients with T2DM patients, and better clinical efficacy is achieved in patients, to varying degrees, by the enrichment of species and the abundance of beneficial bacteria in the gut compared to patients with fewer and less abundant species who experience suboptimal glycemic control. Thus, changes in the intestinal microecological balance can affect the clinical efficacy of patients with T2DM. Diabetic oral liquids (syrups, solutions, beverages, etc.) on the market are mainly composed of a single dietary fiber or a derivative of dietary fiber such as arabinose and lactulose [68]. Preprandial administration in diabetic patients can appropriately stabilize postprandial blood glucose, control obesity, and treat constipation. However, it is difficult to digest and absorb this kind of dietary fiber. A small inadvertent dose can lead to adverse reactions, such as abdominal discomfort, flatulence, anorexia, nausea, vomiting, and diarrhea, and it can even affect the normal physiological function of patients. Therefore, it is important to overcome the negative effects of the abovementioned dietary fiber when used to stabilize blood glucose and lipid levels.

To achieve the above objectives, we developed a new class of prebiotic combination consisting of carbohydrate prebiotics, including lactulose and arabinose, which accounted for 60% of the weight of the total synbiotics, and *Lactobacillus plantarum*, which accounted for 40%.

Lactulose, also known as 4-O-β-D-galactopyranosyl-D-fructose, is a disaccharide consisting of galactose and fructose that is not present in nature and, generally, occurs as a syrupy product. Lactulose has important physiological and pharmacological functions and is widely used in many fields such as clinical medicine, health care products, and food additives. Lactulose is converted in the colon by alimentary tract flora into low molecular weight organic acids, resulting in a decrease in intestinal pH and an increase in fecal volume by retaining water. These effects stimulate colon peristalsis, maintain unobstructed stools, relieve constipation, and restore the physiological rhythm of the colon. In hepatic encephalopathy (PSE), hepatic coma, and precoma, these effects promote the growth of enteric acidophilic bacteria (such as *lactobacilli*), inhibit proteolytic bacteria, and convert ammonia to an ionic state; cathartic effects are exerted by lowering contact pH, exerting osmotic effects, and improving bacterial ammonia metabolism [69,70,71].

L-arabinose can block the metabolic conversion of sucrose, which makes it useful in reducing weight and controlling diabetes. L-arabinose has two main functions in food and pharmaceutical use: One is to inhibit the enzyme that hydrolyzes disaccharides, thus inhibiting the increase in blood glucose caused by the ingestion of sucrose (which is absorbed by intestinal sucrase to break down glucose and fructose); in short, the hypoglycemic effect of inhibiting the hydrolysis of disaccharides. Secondly, due to the inhibitory effect of L-arabinose on disaccharide hydrolase, sucrose that is not decomposed in the small intestine is decomposed by microorganisms in the large intestine to produce a large number of organic acids, which inhibits the synthesis of fat in the liver, which coupled with the inhibitory effect of L-arabinose on sucrose absorption in the small intestine, reduces the production of new fat in vivo [72,73,74].

*Lactobacillus plantarum* has many health functions such as immune regulation; inhibiting pathogenic bacteria; reducing serum cholesterol content and preventing cardiovascular diseases; maintaining the intestinal flora balance; promoting the absorption of nutrients; relieving lactose intolerance; inhibiting the formation of tumor cells. A probiotic combination consisting of lactulose, arabinose, and *Lactobacillus plantarum* is highly effective in controlling blood glucose and lipids as well as body weight levels in patients with T2DM. Lactulose is a strong promoter of *Bifidobacterium* proliferation, which can reduce blood ammonia and plasma endotoxins, and it has the function of “detoxification”; at the same time, lactulose reaches the colon and is decomposed into lactic acid by bacteria, which stimulates intestinal peristalsis and acts as a laxative agent. Arabinose not only has the function of specifically inhibiting sucrase, thus reducing the absorption of glucose and fructose in the intestine of sucrose hydrolysis products to achieve the goal of lowering glucose, it is also a highly effective prebiotic and can regulate the stability and balance of gut microbiota. *Lactobacillus plantarum* can reduce serum cholesterol content and has the function of lowering blood lipids. In addition, it can regulate the immune function of the body, inhibit the growth of pathogenic bacteria, and maintain the balance of intestinal flora. Most crucially, *Lactobacillus plantarum* is superior to other species of lactic acid-lowering lactobacilli in that it is an acid-resistant, alkali-resistant anaerobe and can colonize in large numbers in the intestinal environment, assuring its stable and long-term function [75,76].

With the combined functions of the three probiotics and prebiotics in detoxification, hypoglycemia, and the lowering of lipids, this synbiotic combination can effectively maintain blood glucose and lipid stability in patients with T2DM, effectively avoiding overweight in patients, and it can also eliminate the toxicity and side effects of other single prebiotics on the body. For normal people, this synbiotic combination can play a preventive role and help maintain a healthy weight (Figure 2 and Figure 3). This finding not only provides an important theoretical basis for the mechanism of intestinal microecological regulation affecting the therapeutic effect of diabetes mellitus, but it also develops a new probiotics combination into the dietary regimen for T2DM, providing a new therapeutic assistance for the control of T2DM.

### 4.1. Animal Models

To verify whether the above synbiotic combination was effective in treating T2DM, we investigated the efficacy in diseased mice models. Eight-week-old, SPF-grade male Kunming mice were used in all experiments. After 7 days of adaptive feeding, we randomly selected 10 of these mice as a blank control group. Next, the model for the T2DM mice was established. The mice were fed a high-fat diet (i.e., 10% egg yolk, 10% lard, 1% cholestanol, 0.2% bile salt, and 78.8% basal diet) for one month. After overnight fasting, the model was continued with low-dose alloxan (90 mg/kg, intraperitoneal injection). Two days later, blood was collected from the tail vein to measure the blood glucose level after 8 h of fasting. The mice with blood glucose values ≥ 11 mmol/L were determined as Type 2 diabetes model successfully established experimental animals. Ten of them were randomly selected as the model control groups. The experimental group included a blank control group, model control group, experimental treatment group, and control treatment groups 1–3, with 10 animals in each group, altogether 60 animals. The experiment lasted for three months. The blank experimental group was fed a common diet, and the other groups were fed a high-fat diet. During the experiment, the prebiotic combination powder was taken and prepared as an aqueous solution of prebiotic combination according to a dose equivalent to 10 g/kg, and the dose was 0.2 mL/10 g by intragastric administration every morning. The model control group and the blank control group were given the same volume of normal saline by intragastric administration.

We used different proportions of the synbiotic combination to verify the efficacy on T2DM and finally determined that the following proportions had the best effect and showed the corresponding experimental data. The following are the proportions assigned to the respective synbiotic constituent groups of the experimental and control-treated groups (Table 1).

We selected the four indicators of blood glucose (Glu), total cholesterol (TC), triglyceride (TC), and body weight (Wt) from among eight indicators of body weight change, liver wet weight, fasting blood glucose, insulin, blood lipids, total cholesterol, triglyceride, and alanine aminotransferase, as they can directly reflect the health status of mice. Glu, TC, TG, and Wt were measured before dosing at 4, 8, and 12 weeks of the treatment. The experimental results are shown in Figure 4.

The effect of the synbiotic combination on fasting blood glucose (A); total cholesterol (B); triglycerides (C); body weight (D) in diabetic mice (n = 10). The blank control group was composed of wild-type mice fed a common diet and normal saline; the model control group was composed of T2DM model mice fed a high-fat diet and normal saline; the experimental treatment group and control treatment groups 1–3 were composed of T2DM model mice gavaged with 10 g/Kg of synbiotic aqueous solution. The above data are representative of three independent experiments and presented as the mean ± SEM.

As can be seen from the experimental data in Figure 4, the synbiotic combination consisting of lactulose, arabinose, and *Lactobacillus plantarum* significantly improved the glucose and lipid metabolism in T2DM mice, significantly decreased the blood glucose and lipid levels, and effectively decreased the body weight of mice in the experimental treatment group compared with the model control group. By analysis of the control treatment groups 1–3, we found that the absence of either a prebiotic of lactulose and arabinose or the use of other strains of *Lactobacillus plantarum* affected the efficacy. This synbiotic combination can enrich probiotics to promote their growth and inhibit harmful microbiota, restoring the intestinal microenvironment to a healthier state. Probiotic metabolites can effectively regulate blood glucose, blood lipids, and other metabolic indicators in T2DM mice, ensuring the body maintains a normal metabolism and immune function and improves the clinical therapeutic effect. The synbiotics composed of lactulose, arabinose, and *Lactobacillus plantarum* can not only improve the curative effect of T2DM but also provide the theoretical basis for further the understanding of the molecular mechanism of the interaction between gut microbiota and the metabolic function of the host (CN patent No. ZL 2020108516023).

### 4.2. Clinical Studies

Over the past decade, there has also been an increase in clinical studies on synbiotics in the treatment of T2DM. Probiotics and synbiotics are well-known components in functional and nutritious foods, and the rational selection of synbiotics is beneficial to health, because they affect the ecological stability and immunity of gut microbiota. Selecting a reasonable synbiotic can provide beneficial effects and improve body mass index and fat mass in obese patients. Some synbiotics have been shown to have beneficial effects on insulin receptor substrates (IRS), reducing cell adhesion molecule-1 levels, and reducing insulin resistance and lipid levels. In addition, selected probiotics improved carbohydrate metabolism, fasting glucose, insulin sensitivity, and antioxidant status and reduced metabolic stress in subjects with T2DM [77].

Clinical studies have demonstrated that reasonable synbiotic combinations can reduce body weight and blood pressure in obese patients. The effect of *L. gasseri SBT 2055* was studied in a group of Japanese adults with large visceral fat area (VFA). Participants were randomly assigned to receive increasing numbers of colony-forming units (CFUs) of *L. gasseri SBT2055* for 12 weeks. The results showed a decreased body mass index (BMI), waist circumference, abdominal VFA, and hip circumference [78,79]. There were also relevant studies investigating weight changes in 3724 obese men and women taking *Lactobacillus rhamnosus CGMCC1*, fructooligosaccharide, and inulin over 24 weeks [80]. After 12 weeks, mean weight loss was significantly higher in women in the Lactobacillus rhamnosus group than in women in the placebo group, whereas the mean weight loss was similar in both groups of men. Lactobacillus rhamnosus-induced weight loss in women was associated not only with a significant decrease in fat mass and circulating leptin concentration but also with the relative abundance of *Lacoborferiidae* bacteria in feces; this family belongs to the *Firmicutes*, a taxon previously reported to be positively associated with obesity [54]. These studies suggest that synbiotics can play a good role in the control of obese patients. By controlling body weight, the occurrence of T2DM can be prevented in advance and the condition of patients with T2DM can be controlled.

There is an increasing number of studies on synbiotics in the treatment of T2DM as summarized in Table 2 below.

The above is only a list of the current clinical research progress, while more clinical studies on synbiotics, and synbiotics in the control of T2DM have achieved important results. Through the above clinical studies, we can see that different synbiotic combinations can play a positive role in the treatment of T2DM and effectively control a patient’s condition. Compared with traditional drug therapy, synbiotics therapy is suitable for most patients and has obvious effects, which provides a new possibility for the good of patients with T2DM.

## 5. Regulation Mechanisms of Gut Microbiota by Synbiotics

The adult intestine contains approximately 10^14^ bacterial cells, and approximately 1000 different bacterial species have been found of which approximately 30–40 kinds are common bacteria, accounting for 99% of the total bacterial count. There are more than one hundred bacteria in the normal human intestine, while in patients the number lowers to a several dozen. Various genes in microorganisms are essential for the function of intestinal flora [88]. Microbial ecosystems protect the host from pathogenic microorganisms by competing with the host for nutrition and space, improving the immune system, maintaining intestinal integrity, and biotransformation of drugs [89,90,91,92,93].

Healthy individuals are thought to have a more diverse gut microbiota, because there are fewer species of pathogens that produce vitamins and essential nutrients by degrading complex polysaccharides and maintaining intestinal motility and immune function [94,95], which contributes to the fermentation of undigested food components. Studies have shown that the occurrence of many diseases is related to the imbalance of the intestinal microenvironment [96]. Alterations in the diversity or structure of the gut microbial community, known as imbalances, may affect metabolic activity, leading to obesity and metabolic disturbances, such as metabolic syndrome (MS), dyslipidemia, diabetes, and nonalcoholic fatty liver disease (NAFLD) [97], while synbiotics can improve the intestinal microenvironment and thereby control T2DM. The mechanisms of how synbiotics play a role in the treatment of T2DM are worth studying. Larson et al. found that the ratio of *Bacteroidetes* to *Firmicutes* in the intestine and the ratio of *Bacteroides* province to *Enterococcus* were positively correlated with blood glucose levels in patients with T2DM but not with body mass index. Compared with normal subjects, the number and abundance of deleterious microflora in the stools of diabetic patients increased, with a marked increase in the abundance of beta-proteobacteria [66]. The feces of 345 Chinese patients with T2DM were sequenced using the gut genome sequencing method (MGWAS). The intestinal flora of diabetic patients was moderately disturbed [98,99]. The abundance of harmful microbiota was significantly increased, including opportunistic pathogens and bacteria with sulfate-reducing capacity and antioxidant function, while the number of beneficial microbiota tended to decrease, including the abundance of some butyrate-producing bacteria greatly reduced. Karlsson’s study of intestinal flora in 145 European women with euglycemia, impaired glucose tolerance, and diabetes found that the gut of diabetic patients changed significantly compared with other groups, with the predominance of harmful bacteria, resulting in the disturbance of the intestinal microenvironment and failure of probiotics to perform their original functions [100].

Based on these results the gut microenvironment was altered in patients with T2DM, with a marked reduction in intestinal flora species and a reduced abundance of probiotics leading to the predominance of harmful bacteria. However, it is not known whether alterations in the intestinal microenvironment led to T2DM or metabolic abnormalities in T2DM lead to changes in the intestinal microenvironment.

The current research focuses on the following aspects regarding how gut microbiota participates in regulating the mechanism of T2DM.

(1)The intestinal flora regulates the absorption and utilization of nutrients and energy. The gut is the first gateway for glucose absorption and utilization and plays a crucial role in the regulation of glycemic homeostasis. The gut microbiota can ferment carbohydrates in foods that cannot be digested by the host itself by encoding a large number of glycoside hydrolases to convert them into monosaccharides and broken chain fatty acids (SCFAs), which have been found to alter the composition of the gut microbiota in obese and T2DM patients, affecting the gene expression of broken chain fatty acid receptors and affecting the starvation and repletion cycle of the host [101]. On the other hand, products of intestinal flora (such as methane and SCFAs) can slow intestinal peristalsis, prolong the transit time of intestinal contents, cause enteral nutrition, including glucose, to be more fully absorbed, and directly affect the postprandial blood glucose content. The gut microbiota can also be involved in the pathogenesis of obesity and T2DM by regulating bile acid synthesis and regulating fat and glucose metabolism [102];(2)Intestinal flora is involved in lipogenesis and storage. Significant pathophysiological features of T2DM are insulin resistance accompanied by an absolute or relative deficiency in insulin secretion due to the fact of a defect in pancreatic beta-cell function, and obesity is strongly associated with insulin resistance. Gut microbiota can affect host lipogenesis and storage by a variety of mechanisms. On the one hand, intestinal flora upregulates the expression of the hepatic carbohydrate response element-binding protein and sterol regulatory element-binding protein-1 mRNA, thereby inducing the production of acetyl-CoA carboxylase and fatty acid synthase, key enzymes of lipogenesis, and promoting hepatic triglyceride synthesis. On the other hand, the intestinal flora downregulates fasting-induced adipocytokine (Fiaf) expression produced by intestinal epithelial cells. Fifa inhibits white adipose and muscle tissue from absorbing fatty acids from triglyceride-rich lipoproteins in the blood by acting on lipoprotein lipase (LPL). It was further found that Fiaf can also resist diet-induced obesity by inducing the expression of peroxisome proliferator-activated receptor costimulators, initiating the fatty acid oxidative metabolic pathway, and increasing the transcriptional activity of fatty acid magnesium oxide to increase fatty acid β-oxidation [103];(3)Chronic low-grade inflammatory response are caused by intestinal flora disorder. T2DM has varying degrees of chronic low-grade inflammatory responses characterized by metabolic endotoxemia and disorders of the endocannabinoid system [104]. Available evidence suggests that gut microbiota can affect lipid metabolism and induce systemic chronic low-grade inflammatory responses in animals, leading to the development of obesity and insulin resistance, and this pathogenic role may be much greater than the contribution of animal autogenetic defects to pathogenesis [105].

Researchers have noticed that synbiotics can play a positive role in the treatment of T2DM, but the underlying mechanism has not been clarified. Reports show that synbiotics are fully effective in improving insulin resistance and hyperglycemia in patients with metabolic syndrome and T2DM by controlling intestinal microenvironment homeostasis.

Everard et al. found that prebiotics can improve host health by regulating changes in the intestinal flora related to intestinal endocrine, barrier, and immune function to maintain mucus thickness at a normal level, thus improving glucose and lipid metabolism in patients. A high-sugar diet and low dietary fiber intake lead to an abnormal intestinal flora composition, alter intestinal permeability, and inflammatory factors enter the blood circulation through the intestinal barrier to elicit an inflammatory response, which is precisely improved by prebiotics, and keep the intestine in a healthy state [106]. In addition, it has been found that the treatment with metformin, a widely used hypoglycemic drug, can significantly increase the number of goblet cells and the level of *Akkermansia* (degradable mucin) in the intestine of obese mice fed with a high-fat diet [107].

In 2017, Gu Yanyun et al. first conducted a study on the relationship between the efficacy of antidiabetic drugs and the characteristics of intestinal symbiont microbiota, which not only solved the mystery of the mechanism of metabolic benefit beyond the hypoglycemic effect of acarbose but also provided new research ideas for the design of novel diabetic drugs targeting bile acid metabolism by intestinal symbionts [108]. The researchers typed the intestinal commensal flora of patients with newly diagnosed T2DM before administration, yielding two different types: Bacteroides-rich enteric and Prevotella-rich enteric types, and found that patients with Bacteroides endosymbionts were more effective after treatment without significant differences in baseline characteristics between the two groups, suggesting that intestinal symbiont characteristics may be an important factor affecting human response to T2DM drugs.

In addition, a study by Zhao Liping’s team found that by providing rich and diverse dietary fiber, the specific beneficial microbiota in the human gut can be increased, thereby improving the clinical symptoms of T2DM [109]. Increasing a large variety of dietary fiber can significantly improve insulin secretion and insulin sensitivity in patients with T2DM by altering the structure of the microbiota; at the strain level, a group of “short-chain fatty acid”-producing bacteria, which are conducive to increasing insulin secretion and insulin sensitivity, can be regarded as “ecological functional groups” necessary for the restoration and maintenance of human health. The higher the abundance and diversity of these “flora” bacteria are restored, the lowering of HbA1c decreases. A reasonably designed high-fiber diet specifically promotes the growth of these “flora” bacteria for each patient’s microbiota characteristics or may be a new approach to personalized nutritional therapy for diabetes in the future.

Through the above cases, we can control the abundance of intestinal flora or supplement new strains through the addition of different kinds of synbiotics to obtain a better effect in the treatment of T2DM. The mechanisms of intestinal microecological regulation by synbiotics is now starting to emerge, and the mechanisms of beneficial bacteria in the intestine has been more thoroughly explored, which can effectively promote the study of synbiotics in the treatment of T2DM.

## 6. Future Perspectives

More attention has been given to the study of synbiotics in the treatment of T2DM. The regulation of gut microbiota through synbiotics is a potential mechanism. According to the different characteristics of synbiotics, it is theoretically feasible to select different and appropriate combinations to regulate intestinal microbial homeostasis in patients with T2DM, which is theoretically beneficial and has been confirmed by many studies. Along with the study of the relationship between gut microbiota, obesity and T2DM, some researchers have suggested that intervention of gut microbiota may be one of the possible means in the field of diabetes prevention and treatment in the future.

Synthetic bacteria can modulate the immune system by producing short-chain fatty acids, thereby improving glucose homeostasis [29,77,110,111,112,113]. However, high doses of synbiotics consumption may cause discomfort, such as bloating and flatulence, due to increased production and fermentation of short-chain fatty acids [29]. Therefore, we need to control the dosage of synbiotics and the appropriate proportion of probiotics and prebiotics. In our study, we found a reasonable synbiotic ratio and achieved a promising therapeutic effect: the prebiotic combination in this study can significantly improve glucose and lipid metabolism, significantly reduce blood glucose and lipid levels in T2DM mice and can effectively reduce the body weight. Prebiotic combinations—lactulose and arabinose—and *Lactobacillus plantarum* promote the enrichment and growth of probiotics in the intestine and inhibit the retention and accumulation of harmful bacteria by regulating the balance of gut microbiota. Lactulose can reduce blood ammonia and plasma endotoxins to stimulate intestinal motility; arabinose can reduce blood sugar and regulate gut microbiota; while metabolites of *Lactobacillus plantarum* can effectively regulate blood glucose, blood lipids, and other metabolic indexes in T2DM mice, ensuring that the body maintains a normal metabolism and immune function and improves the clinical therapeutic effect. We hope to use this combination of prebiotics in the clinic to control blood glucose and lipid levels, maintain weight, and reduce the side effects in patients with T2DM. We can also use synbiotics to control weight and maintain health in normal people.

The prevalence of T2DM is increasing, with an average of 1 in 10 people worldwide, especially in countries with an unreasonable diet, and we must pay attention to the fact that early prevention is the most effective and safest way to control it. New technologies, such as 16S rRNA, gradient gel electrophoresis based on polymerase chain reaction denaturation, metagenomics, metatranscriptomics, and microarrays, have contributed to the exploration of a wide range of gut microbiota [114,115]. It urges us to study the potential linkage between gut microbiota and T2DM. Further understanding of the mechanism between them will provide new treatments from the intestinal microenvironment or intervene in the development of T2DM in advance by regulating intestinal microenvironment homeostasis. Synbiotics can regulate intestinal flora to maintain homeostasis, providing a new solution for the treatment of T2DM. We expect that synbiotics can become a successful method for curing T2DM.

## Figures and Tables

**Figure 1 foods-11-02438-f001:**
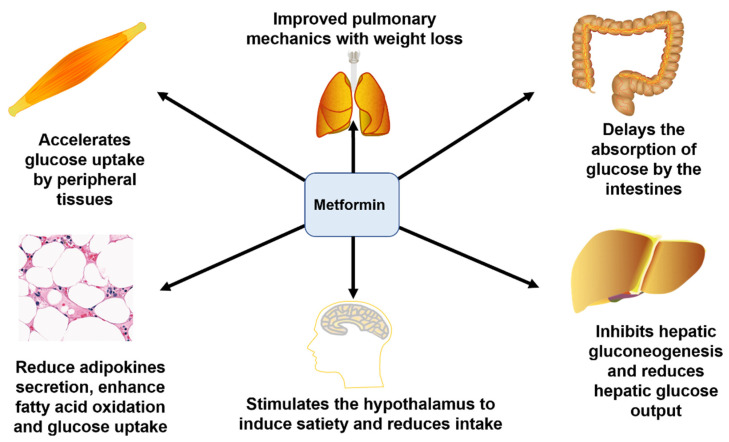
Mechanism of the function of metformin in the treatment of T2DM.

**Figure 2 foods-11-02438-f002:**
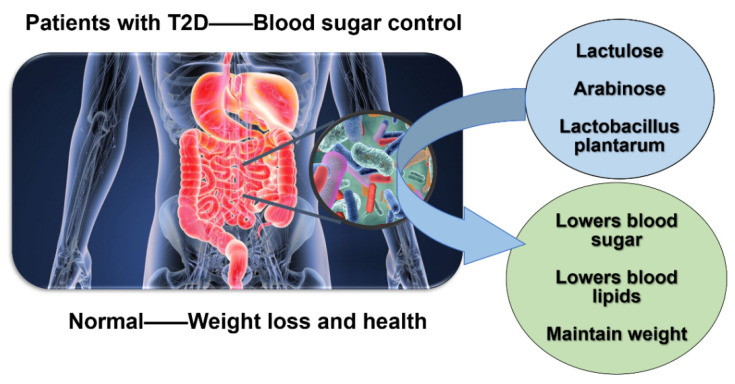
Function of the new combination of probiotics.

**Figure 3 foods-11-02438-f003:**
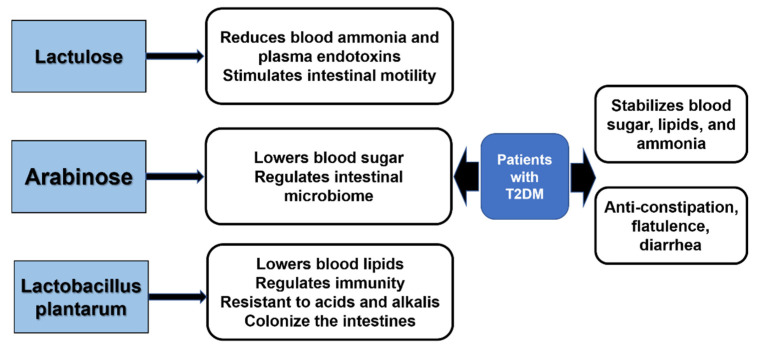
Effect of lactulose, arabinose, and *Lactobacillus plantarum* in the treatment of T2DM.

**Figure 4 foods-11-02438-f004:**
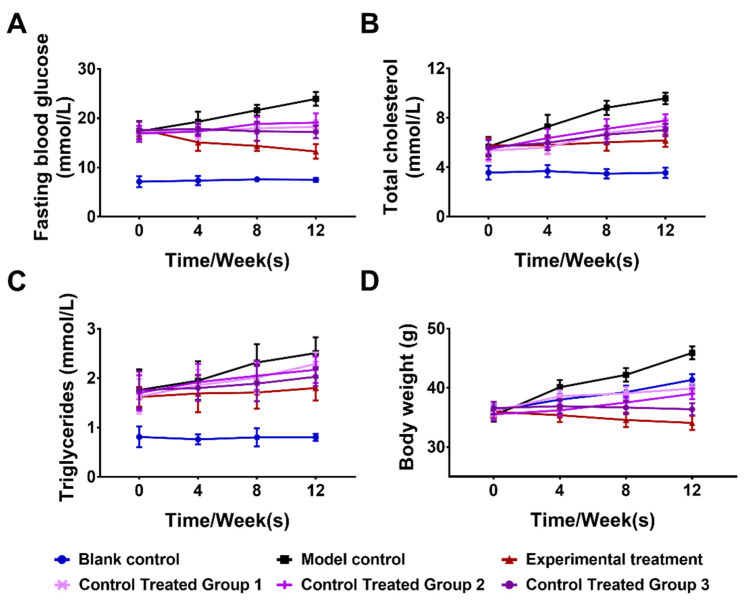
Effects of the synbiotic combination on the sugar and fat metabolism of mice.

**Table 1 foods-11-02438-t001:** The proportion of synbiotic components in each experimental group.

Experimental Group Assignment	Symbiotic Components		
Experimental treatment groups	30% Lactulose	30% Arabinose	40% *Lactobacillus plantarum CGMCC 8198*
Control Treated Group 1	60% Lactulose	\	40% *Lactobacillus plantarum CGMCC 8198*
Control Treatment Group 2	\	60% arabinose	40% *Lactobacillus plantarum CGMCC 8198*
Control Treatment Group 3	30% Lactulose	30% Arabinose	40% *Lactobacillus plantarum CGMCC 1258*

**Table 2 foods-11-02438-t002:** Clinical studies of different synbiotics in the treatment of T2DM in recent years.

Reference	Sample	Strain/Dose	Time	Results
Moroti et al., 2012 [81]	20 patients with T2DM	*B. bifidum* 10^8^ CFU, L. acidophilus 10^8^ CFU, and 2 g oligofructose	2 weeks	Increased HDL-C and reduced fasting glycemia.
Asemi et al., 2013 [82]	54 patients with T2DM	*L. acidophilus*, *L. casei, L.rhamnosus, L.bulgaricus, B. breve, B. longum, S. thermophilus*, 10^9^ CFU and 100 mg FOS	8 weeks	TGL and HOMA-IR plasma levels increased; serum CRP decreased.
Tajadadi-Ebrahimi et al., 2014 [83]	81 patients with T2DM	*L. sporogenes*, 10^8^ CFU and 0.07 g inulin per 1 g	8 weeks	Reduce serum insulin levels; conducive to insulin metabolism.
Shakeri et al., 2014 [84]	78 patients with T2DM	*L. sporogenes*, 10^8^ CFU and 0.07 g inulin per 1 g	8 weeks	The serum HDL-C level significantly increased; the blood lipid profile decreased (TAG, TC/HDL-C).
Nazila Kassaian et al., 2016 [85]	120 adults with impaired glucose tolerance	*Lactobacillus acidophilus, Bifidobacter bifidum, Bifidobacter lactis,* and *Bifidobacter longum* (10^9^ CFU) with maltodextrin as filler and 6 g inulin	6 months	Elevated HDL-C, and improved (LDL)/HDL.
Hossein et al., 2019 [30]	136 patients with T2DM	*Lactobacillus acidophilus* 10^8^ CFU and 0.5 g of powdered cinnamon	3 months	Improved antioxidant enzyme activity modestly.
Soleimani et al., 2019 [86]	60 patients with diabetes mellitus complicated with hemodialysis	*Lactobacillus acidophilus, Lactobacillus casei, Bifidobacterium bifidum* (2 × 10^9^ CFU) and 0.8 g inulin	12 weeks	Reduced blood glucose, insulin levels, and insulin resistance; improved insulin sensitivity.
Aynaz Velayati et al., 2021 [87]	50 patients with T2DM	*Bacillus Coagulans, Lactobacillus rhamnosus, Lactobacillus acidophilus* and fructooligosaccharide	12 weeks	Reduced insulin level, HOMA-IR CRP, and HOMA-β levels.

CFU, colony-forming unit; TGLs, total glutathione levels; CRPs, C-reactive protein; FOS, fructooligossacharides; HDL-C, high-density lipoprotein cholesterol; HOMA-IR, homeostasis model assessment of insulin resistance; HOMA-β, β-cell function; LDL-C, low-density lipoprotein cholesterol; T2DM, type 2 diabetes mellitus.

## Data Availability

The data are available from the corresponding author.

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
