# Peer review of "Synbiotics and Gut Microbiota: New Perspectives in the Treatment of Type 2 Diabetes Mellitus"

_foods, 2022, doi:10.3390/foods11162438_

Round 1

Reviewer 1 Report

Your manuscript is important of the field study of T2DM and I appreciate the opportunity to evaluate your work. The use of animal model, clinical studies and despite being necessary caution when extrapolating the results from the animal model to patients, the findings are important to unravel the changes involved in the use of the synbiotic and gut microbiota. Please find attached my comments about the work (mainly the lines: 41, 127, 342, 365, 374, 458, 593, 629-631).

Reviewer 2 Report

Major issues:

The manuscript is poorly written and scientific soundness is missing. Several sentences were modified using the synonym option of MS word.

The abstract is not informative and needs extensive revision

The history of T2DM treatment: it is like a story; this part should be rewritten in a scientific way.

What are the limitations of metformin in DM treatment?

2.2 Function of synbiotics: This section is not informative and failed to detail the intended meaning.

2.3 Study on synbiotics: what you are trying to explain in this section?

The figures are not scientific and informative.

3.1. Animal models. Is this original research data? If so, it is irrelevant, part with insufficient scope. No experimental details and no information about the methodology. No discussion.

The clinical studies part needs to revise for better insight.

 Some of the minor errors:

Line 23. Change Per to every

Line 24. DM is already a chronic disease.

Line 34, 40 & 133. What do you mean by “traditionally”

Line 38. What variant?

Line 43. Add reference.

Line 38 to 52: there are no references for these statements. The sentences are noninformative.

Line 54. Remove “commonly found in the elderly ones”

Line 54-60. References are missing.

Line 64. What do you mean by “attracted”

Line 72. What do you mean by “acids as raw materials”

Line 72. Gospel?

Line 76-78. Patients are not divided into two groups. It’s wrong information.

Line 106-110. Need references

Line 134. What we imagined

Line 146-155. No evidence/references.

Line 146-149, 165-169. The definition is not scientifically approved. Use approved definition.

Lin 181. “The colon is the place where synbiotics work.” Is this correct? References?

Reviewer 3 Report

This is a comprehensive review with a lot of useful information.

Some minor points:

The current treatment for T2DM is known. There is no novel information in section 1.

The switching from section 1 to section 2 is not smooth as the authors state no reason why synbiotics is discussed here.

How to define the dosages for the treatments in line 365? Are the animal models (age, sex etc) and the treatment regimens the same for the studies in Tables 1-4? Are the date in Tables 1-4 done by the authors, or references are needed. Why are these 4 studies chosen for this review?

Round 2

Reviewer 2 Report

The manuscript has been improved